# Network pharmacology and integrative bioinformatics analyses identify PDE1A as a key target of pirfenidone in idiopathic pulmonary fibrosis

Jing Wu[1,2], Haseeb Khaliq[3], Yanyan Ke[1,4]*, Qudrat Ullah [5]*, Sheikh Arslan Sehgal[6], Xue Yi[1,4]*

1 Key Laboratory of Functional and Clinical Translational Medicine, Fujian Provincial Department of Education, Xiamen Medical College, Xiamen, China, 2 Shenzhen Center for Disease control and prevention, Shenzhen, Guangzhou, China, 3 Department of Anatomy and Histology, Cholistan University of Veterinary and Animal Sciences, Bahawalpur, Pakistan, 4 Institute of Respiratory Research, Xiamen Medical College, Xiamen, Fujian, China, 5 Department of Theriogenology, Cholistan University of Veterinary and Animal Sciences, Bahawalpur, Pakistan, 6 Department of Genomics and Bioinformatics, Cholistan University of Veterinary and Animal Sciences, Bahawalpur, Pakistan

* gillke@163.com (YK); qudratullah1@cuvas.edu.pk (QU); yxue@xmmc.edu.cn (XY)

## Abstract

Pirfenidone, an antifibrotic agent, has been shown to be effective in the treatment of idiopathic pulmonary fibrosis (IPF). However, the exact mechanism of action and clinical efficacy require further investigation and validation. This study commenced by identifying pathogenic genes associated with IPF through the GeneCards database. Potential targets of pirfenidone were subsequently screened through PubChem and Swiss TargetPrediction, and overlapping targets were identified through Venn diagram analysis. Enrichment analysis of potential target genes was performed to identify the key biological processes and pathways involved in the action of pirfenidone. The main target genes were subsequently identified through the GSE10667 and GSE110147 datasets. The affinity of PDE1A to pirfenidone was predicted by molecular docking and MicroScale Thermophoresis (MST). Finally, the expression and antifibrotic effects of pirfenidone on PDE1A were validated through data from the GSE226249 dataset. PDE1A, identified by GeneCards and Swiss TargetPrediction, was found to be an important mediator of the antifibrotic effect of pirfenidone. The enrichment analysis revealed biological processes such as cyclic nucleotide-mediated signaling and cAMP-mediated signaling. KEGG pathway analysis further linked pirfenidone activity to pathways involved in calcium signaling, taste transduction, morphine dependence, renin secretion and purine metabolism. Molecular docking, molecular dynamics (MD) simulations and MST results revealed a strong binding affinity between pirfenidone and PDE1A. MD simulations showed the stability of the complex. It was observed that the RMSD analysis of the complex stabilized between 0.6 to 0.8 nm throughout the simulation, however RMSF showed minimal fluctuation.

**Data availability statement:** All trajectory data and the code used to generate the figures and results presented in this study are publicly available in the Zenodo repository at https://doi.org/10.5281/zenodo.18387883.

**Funding:** This work was supported by the Natural Science Foundation of Fujian Province of China (Project Number：2022J011409, 2023J011653). This work was also supported by the Xiamen Medical College Interdisciplinary Innovation Group Project in China (Project Number: K2023-07). Xiamen Science and Technology Bureau Medical and Health Guidance Project (Project Number: 3502Z20199136). Fujian Province Joint Fund for Scientific and Technological Innovation Project (2024Y9728). The above-mentioned grants were jointly received by the following authors: Jing Wu, Yanyan Ke, and Xue Yi.

**Competing interests:** The authors have declared that no competing interests exist.

Data from the GSE226249 dataset confirmed that upregulation of PDE1A promotes fibrosis, whereas pirfenidone downregulates PDE1A, thereby exerting its antifibrotic effect. The inhibition of IPF progression by pirfenidone is mediated by PDE1A, providing insights into its therapeutic mechanism.

## Introduction

IPF is a chronic and progressive interstitial lung disease characterized by the destruction of alveolar structures and irreversible fibrotic remodeling, leading to severe deterioration in patients' quality of life and a poor prognosis [1–3]. Although research into the pathogenesis of IPF has made some progress in recent years, current clinical treatment options remain limited, and their efficacy is suboptimal. Therefore, the exploration of new therapeutic targets and pharmacological intervention strategies is urgently needed. Pirfenidone, an antifibrotic drug approved for the treatment of IPF [4,5], has been shown to slow the progression of fibrosis by modulating several cellular and molecular pathways involved in fibrogenesis. Its mechanisms of action include anti-inflammatory, antioxidant, and antiproliferative effects. However, the exact mechanism by which pirfenidone exerts its antifibrotic effect is not yet fully understood.

PDE1A is an intracellular enzyme responsible for the degradation of cyclic adenosine monophosphate (cAMP) and cyclic guanosine monophosphate (cGMP), two important intracellular messengers involved in the regulation of various cellular functions and biological processes, such as proliferation, cell differentiation, migration, and inflammation. In the context of IPF, increased PDE1A activity may lead to decreased cAMP and cGMP levels, thus promoting fibroblast activation and fibrosis [6,7]. In this study, we used a comprehensive approach combining pharmacology and bioinformatics [8–10] to investigate the key mechanisms by which pirfenidone treats IPF. Specifically, we focused on the regulation of PDE1A by pirfenidone. By building drug–target disease networks and performing molecular docking analyses, we aimed to clarify how pirfenidone disrupts the fibrotic process in IPF by targeting PDE1A. The results of this study provide new insights into the mechanism of action of pirfenidone and may provide a theoretical and experimental basis for the personalized treatment of IPF. By further investigating the effect of pirfenidone on PDE1A, we aimed to advance the development of innovative therapeutic strategies for IPF, with the goals of improving patients' quality of life and improving their prognosis.

## Materials and methods

### Identification and analysis of differentially expressed genes in IPF

GeneCards database (https://www.genecards.org/) was used to retrieve the disease-related genes by using the keyword "IPF" and 6,896 IPF targets were identified. Additionally, the datasets GSE10667, GSE110147, and GSE226249 from the NCBI GEO database (http://www.ncbi.nlm.nih.gov/geo/) was also retrieved. The following sample groups were used in this study: for the GSE10667 dataset, the normal samples included GSM269749--GSM269763, while the IPF model samples included GSM373881--GSM373888. For the GSE110147 dataset, the normal

samples were GSM29787889-GSM2978799, and the IPF model samples were GSM2978752-GSM2978773. For the GSE226249 dataset, the normal samples were GSM7068937-GSM7068939, the IPF samples were GSM7068946-GSM7068948 and GSM7068955-GSM7068957, and the IPF+ pirfenidone samples included GSM7068949-GSM7068950 and GSM7068958-GSM7068960.

## Identification of pirfenidone targets

To identify the potential targets of pirfenidone, the chemical information of the identified targets was retrieved from PubChem website (https://pubchem.ncbi.nlm.nih.gov/) and obtained its SMILES structures. The SMILES string was submitted to the SwissTargetPrediction platform (http://swisstargetprediction.ch/), with "*Homo sapiens*" selected as the target species. The default probability-based prediction model of SwissTargetPrediction 2024 were used and a total of 101 putative pirfenidone-associated targets were obtained. All the predicted targets were exported directly and corresponding probability scores were recorded to ensure the reproducibility. Total 6896-assoociated genes for the disease-related targets were retrieved from the GeneCards database by using default scoring system.

The overlap between IPF-related genes (6896) and pirfenidone-associated targets (101) was determined by using the Venn diagram. 71 shared targets were identified, which were considered the potential therapeutic targets through which pirfenidone may exert its effect in IPF.

## Functional enrichment analysis

Functional enrichment analyses of the identified 71 overlapping targets was performed by utilizing Metascape (https://metascape.org/gp/index.html#/main/step1) for protein-protein interaction (PPI) analyses, pathway enrichment and gene annotation. All 71 DEGs were subjected to Metascape platform and enrichment analysis was performed by using the default KEGG, GO molecular Function (MF), Reactome, GO Cellular Component (CC) and GO Biological processes (BP) annotation systems. Enrichment score > 1.5, minimum overlap ≥ 3 genes and Benjamini-Hochberg correction for multiple testing were set as statistical threshold. Moreover, all human genes were set as background set. A functional enrichment analysis was performed on the differentially expressed genes to identify relevant biological processes and pathways.

## The human protein atlas

The expression profile of PDE1A across lung-related cell types was obtained from the Human Protein Atlas (HPA) database (https://www.proteinatlas.org/) using the Tissue Expression and Single Cell Type modules. The corresponding normalized expression matrices were downloaded and used to validate the PDE1A expression in lung fibroblasts.

## Molecular docking

The binding interaction between PDE1A and pirfenidone was investigated using AutoDock Vina 1.2.5 employing ssemi-flexible docking, where the ligand (pirfenidone) was treated as flexible and the receptor (PDE1A) as rigid. No experimental PDE1A crystal structure was available, the 3D protein structure was predicted using homology modeling, threading and *ab initio.* The predicted structures were evaluated and the structure predicted through AlphaFold3 (UniProt ID: P5475) was selected for further analyses. The 3D structure of pirfenidone (PubChem CID: 40632) was downloaded from the PubChem d and energy-minimized using MMFF94 force-field minimization.

The compound was reconstructed manually using StoneMIND Collector, followed by 3D optimization. The active binding pockets of PDE1A were predicted using KVFinder [11], with parameters integrating UniProt catalytic annotations. Molecular docking was performed using a grid box fully covering the predicted catalytic pocket. Least binding energy, clustering consistency and highest binding affinity were set for pose selection. The docked complexes were visualized and analyzed using Discovery Studio Visualizer.

## Molecular dynamics simulations

Post docking analyses were performed using molecular dynamic (MD) simulation in GROMACS. The top-ranked selected compound was simulated for 200 nanoseconds (ns). Optimization and minimization of the receptor-ligand complex were carried out. The system was prepared with the Transferable Intermolecular Interaction Potential 3 Points (TIP3P) solvent model in an orthorhombic box. CHARMM36 force field was applied. Counter-ions were added to neutralize the model. The simulation was performed under an NPT ensemble at 303 K temperature and 1 atm pressure. The system was relaxed prior to MD simulation. Trajectories were recorded every 100 ps, and the stability was verified by analyzing the root mean square deviation (RMSD) of both protein and ligand over time.

## MicroScale Thermophoresis (MST) assay

To experimentally validate PDE1A–pirfenidone binding, MicroScale Thermophoresis (MST) analysis was performed using PDE1A (Cat: P89-30H, SinoBiological, China) and pirfenidone1A (Cat: S2907, Selleck, USA).

**Protein labeling.** A total of 100 nM RED-tris-NTA dye was prepared by mixing 2 μL of dye with 98 μL of PBS-T, followed by the addition of 100 μL of 200 nM PDE1A protein. The mixture was incubated at room temperature for 30 minutes in the dark. After incubation, centrifugation was performed at 15,000 × g for 10 minutes at 4 °C, and the supernatant was collected for analysis.

**Fluorescence intensity detection.** Following labeling, 200 μL of protein solution was obtained. A 10 μL aliquot was diluted threefold with assay buffer. The sample was transferred into capillaries and analyzed using the MST instrument tray. Instrument settings were configured as follows: excitation power set to Auto, MST power to Medium, and appropriate capillary type selected.

**Binding affinity measurement.** Binding affinity was quantified by preparing a two-fold dilution series of pirfenidone in assay buffer. Starting with the highest concentration, 20 μL was placed in tube 1, followed by serial 10 μL transfers into tubes 2–16, each containing 10 μL buffer. 10 μL of labeled protein was added to each tube, mixed, incubated for 15 minutes at room temperature and loaded into capillaries. Measurement was acquired using the MST software, with three biological replicates to ensure reproducibility.

## Validation of PDE1A expression in the GSE10667, GSE110147, and GSE226249 datasets

PDE1A expression was validated using the GSE10667, GSE110147 and GSE226249 datasets. Raw data were processed. ROC curve analysis (GraphPad Prism 8.0) was performed for PDE1A in GSE10667 and GSE110147 to evaluate diagonostic performance. Different expression within GSE226249 further validated PDE1A expression pattern.

## Statistical analysis

Quantitative data were expressed as means ± standard deviations (SD) after confirming normality. For normally distributed data, group comparisons were made using one-way ANOVA, followed by Tukey's post hoc test. GraphPad Prism 8.0 was used for all the statistical analyses, with $P < 0.05$ considered statistically significant.

## Results

### Identification of potential targets of pirfenidone in IPF

The research workflow presented in Fig 1 outlines the identification of key targets for pirfenidone in the treatment of IPF. A total of 6,867 IPF-related genes were identified through the GeneCards database. Simultaneously, 101 potential drug targets for pirfenidone were identified from the Swiss TargetPrediction website. Overlapping these datasets through a Venn diagram yielded 71 overlapping genes, which were postulated as potential therapeutic targets for IPF (Fig 2A-2B). Functional enrichment analysis of these 71 potential target genes, performed through Metascape, revealed their involvement

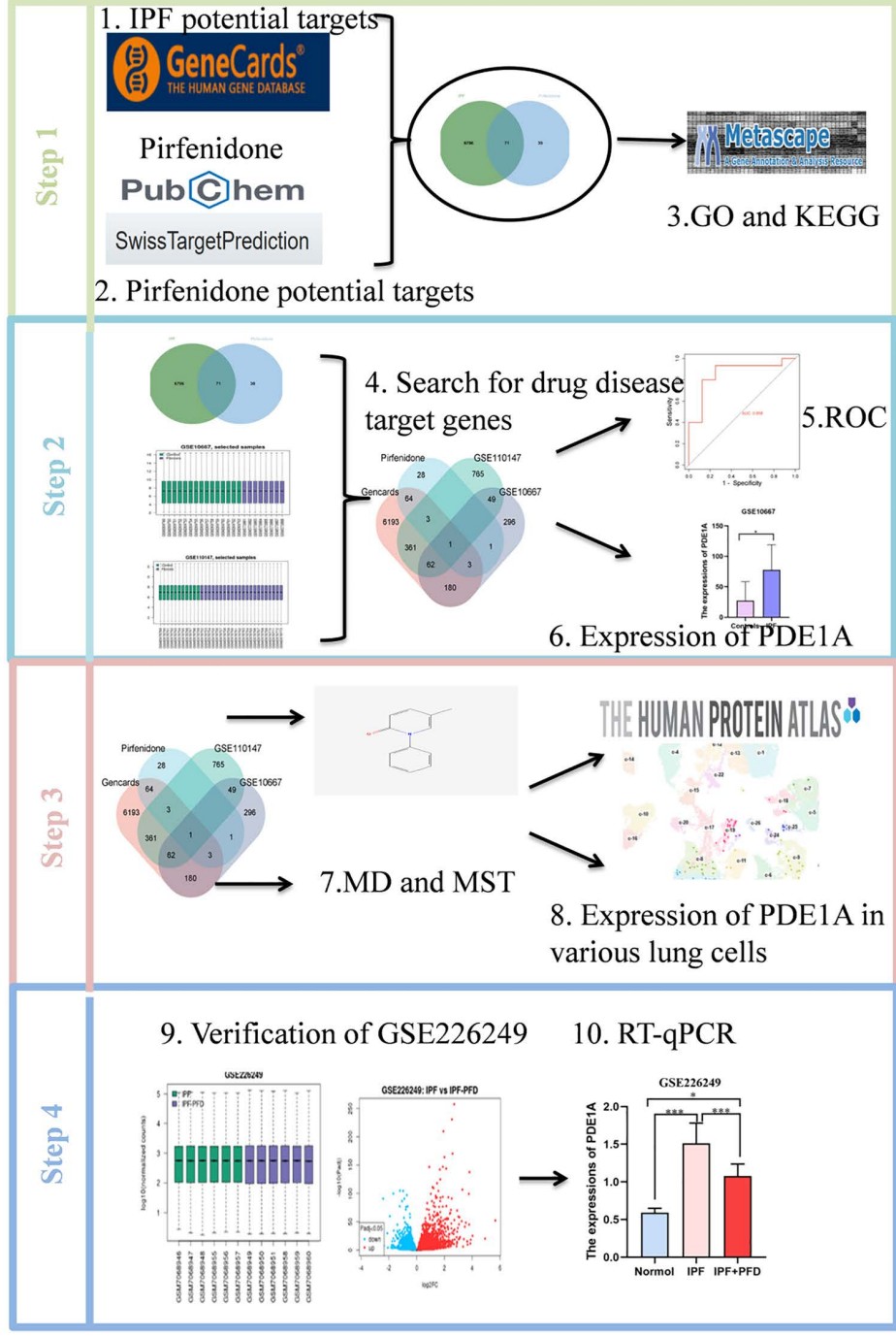

**Fig 1. Flowchart.** The research workflow outlines the process of identifying key targets for pirfenidone in treating Idiopathic Pulmonary Fibrosis.

in several biological processes, including cyclic nucleotide-mediated signaling pathways and cAMP-mediated signaling pathways. In addition, KEGG pathway analysis revealed significant associations with calcium signaling, taste transduction, morphine dependence, renin secretion, and purine metabolism (Fig 2C).

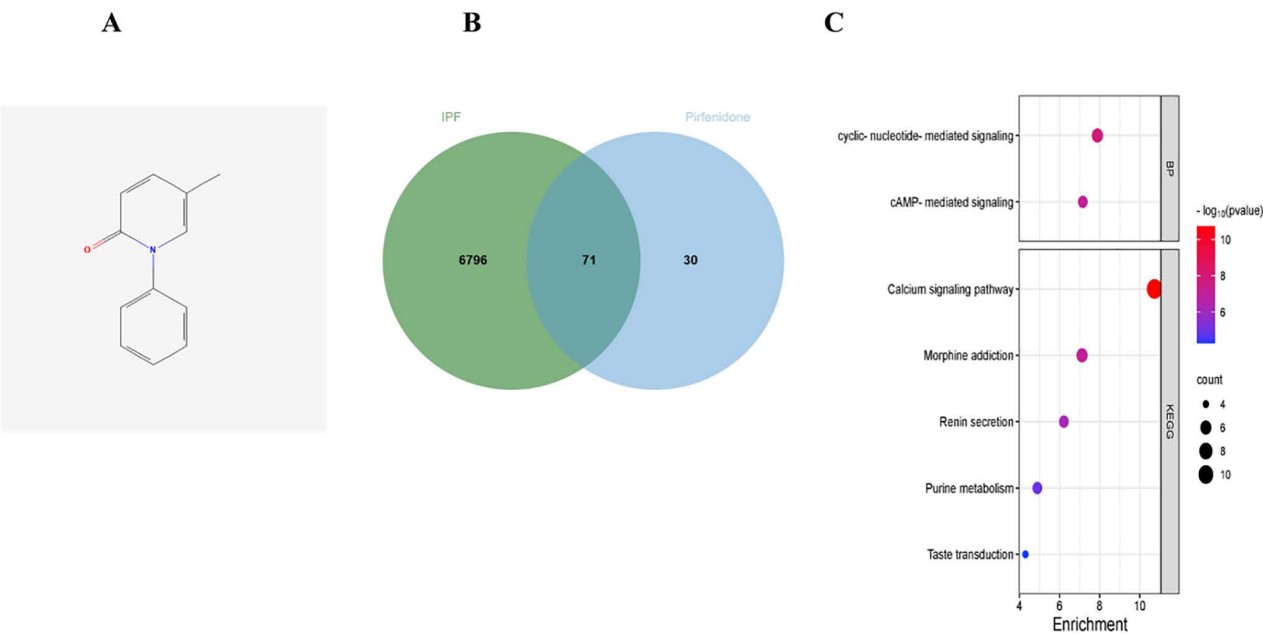

**Fig 2. Screening of differential genes.** A: Molecular structure of pirfenidone; B: Vento screened the potential genes of pirfenidone for IPF treatment; C: Metascape enrichment analysis of potential genes of pirfenidone for IPF treatment.

## Screening of differential genes and core target identification

Differentially expressed genes associated with IPF were identified through the GSE10667 and GSE110147 datasets together with IPF-related genes from the GeneCards database. When the drug targets of pirfenidone from the Swiss TargetPrediction website were combined, PDE1A emerged as a major target in the treatment of IPF (Fig 3A-3E).

## Validation of the core target gene PDE1A in IPF

PDE1A expression was confirmed in several datasets. In the GSE10667 dataset, PDE1A expression was significantly greater in the IPF group than in the control group (Fig 4A). ROC curve analysis further confirmed the diagnostic power of PDE1A, with high AUC values supporting its efficacy in identifying IPF (Fig 4B). Similarly, the GSE110147 dataset revealed increased PDE1A expression in IPF samples, and the ROC curve reflected strong diagnostic performance (Fig 4C-4D). PDE1A is highly expressed in fibroblasts, smooth muscle cells and endothelial cells in lung tissue and is critical for the pathophysiology of pulmonary fibrosis (Fig 4E).

## Interaction analysis of Pirfenidone and PDE1A: MD and MST

The binding patterns of PDE1A (Fig 5A) and pirfenidone (Fig 3B) were predicted through molecular docking (Fig 5C). The docking results indicated that the binding energy between PDE1A and pirfenidone was −6.4 kcal/mol (Table 1), and they formed extensive interactions. Specifically, the side chains of amino acids H219 and H263 of PDE1A formed hydrogen bonds with pirfenidone. In addition, ligands also have π-π stacking and π- cation interactions with H219 residues. These bonding and non-bonding interactions work in synergy, enhancing the binding stability of the complex (Fig 5D).

In this study, we conducted a detailed analysis of the interaction between Pirfenidone and PDE1A through Microscale Thermophoresis (MST) technology. The experimental conditions were as follows: Ligand Concentration: 1000 μM to 0.0305 μM; Excitation Power: 100%; MST Power: 40%; Temperature: 25.0°C; Kd: 1.5097E-05; Kd

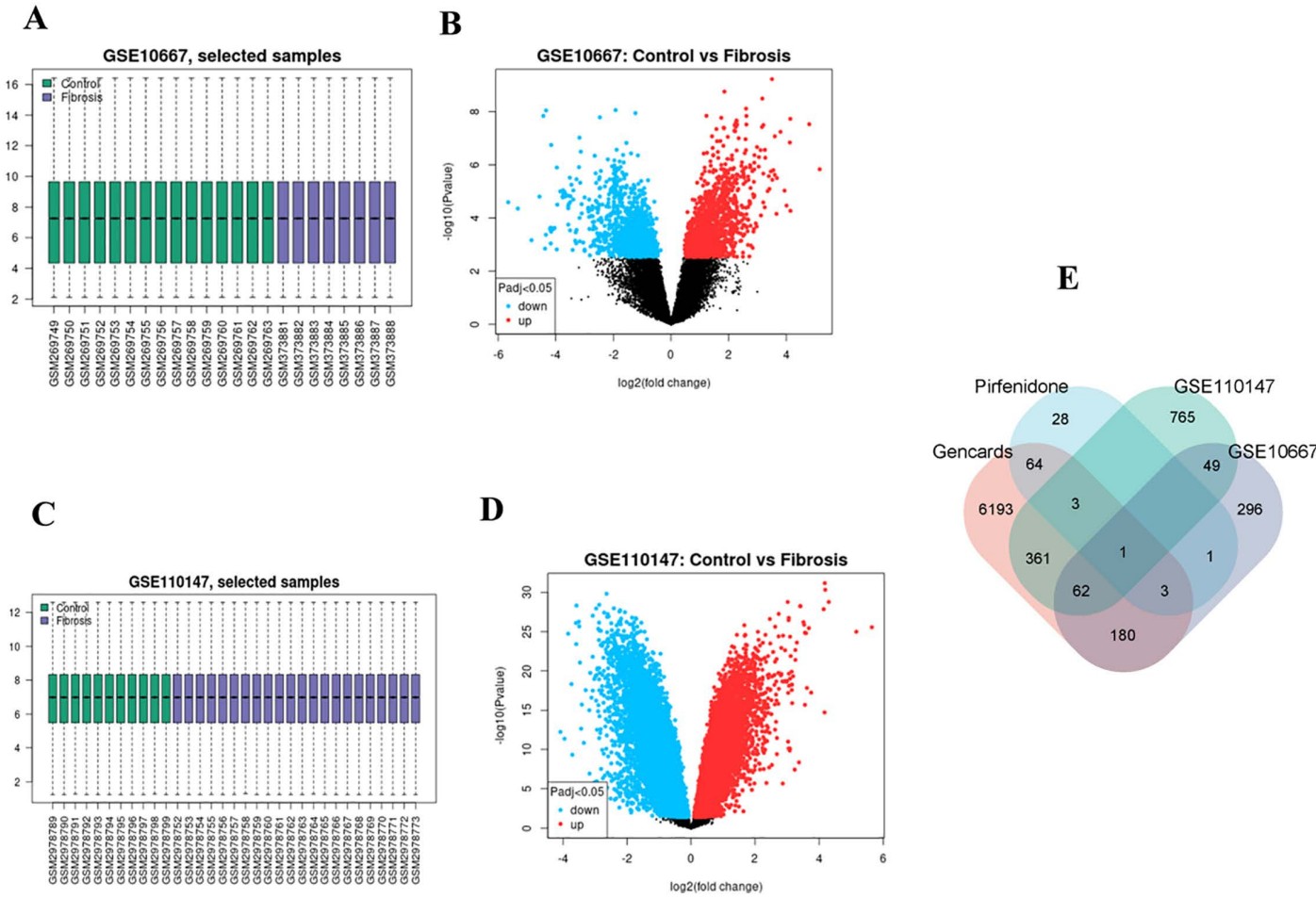

**Fig 3. Screening of core target genes.** A: GSE10667 data set box diagram; B: GSE10667 data set volcano map; C: GSE110147 data set box diagram; D; GSE110147 data set volcano map; E Vento screening the core targets of pirfenidone treatment for IPF.

Confidence: ± 2.5858E-06; Signal to Noise: 19.171163. The MST results showed that a signal-to-noise ratio greater than 5 is acceptable, while valued above 12 indicates high-quality data. The protein mobility slowed with increasing analyte concentration. The fitting curve was S-shaped, exhibiting clear upper and lower plateau phases and strong concentration dependence; indicated specific binding (Fig 5E-5F). The dissociation constant (Kd = 15.1 μM) indicated relatively strong binding affinity *in vitro*.

Using Discovery Studio Visualizer, hydrogen bond interactions were identified between His219 and His263 of PDE1A and the small molecule, while His219 also formed Pi-Pi and Pi-cation interactions with ligand. Van der Waals forces contributed by surrounding amino acids stabilized the ligand within the binding pocket (Fig 6A). The first frame of the dynamic simulation trajectory was used as a reference to calculate the root mean square deviation (RMSD) of Cα atoms. In the initial stage (0–20 ns), the RMSD of both systems increased rapidly, indicating structural adjustment towards equilibrium. Subsequently, the RMSD fluctuation range of the complex stabilized around 0.6–0.8 nm, showing a stable trend, with minor fluctuations. The protein monomer exhibited slightly wider RMSD fluctuation, approximately 0.6–1.0 nm, and an upward trend in the later stage of the simulation (160–200 ns).

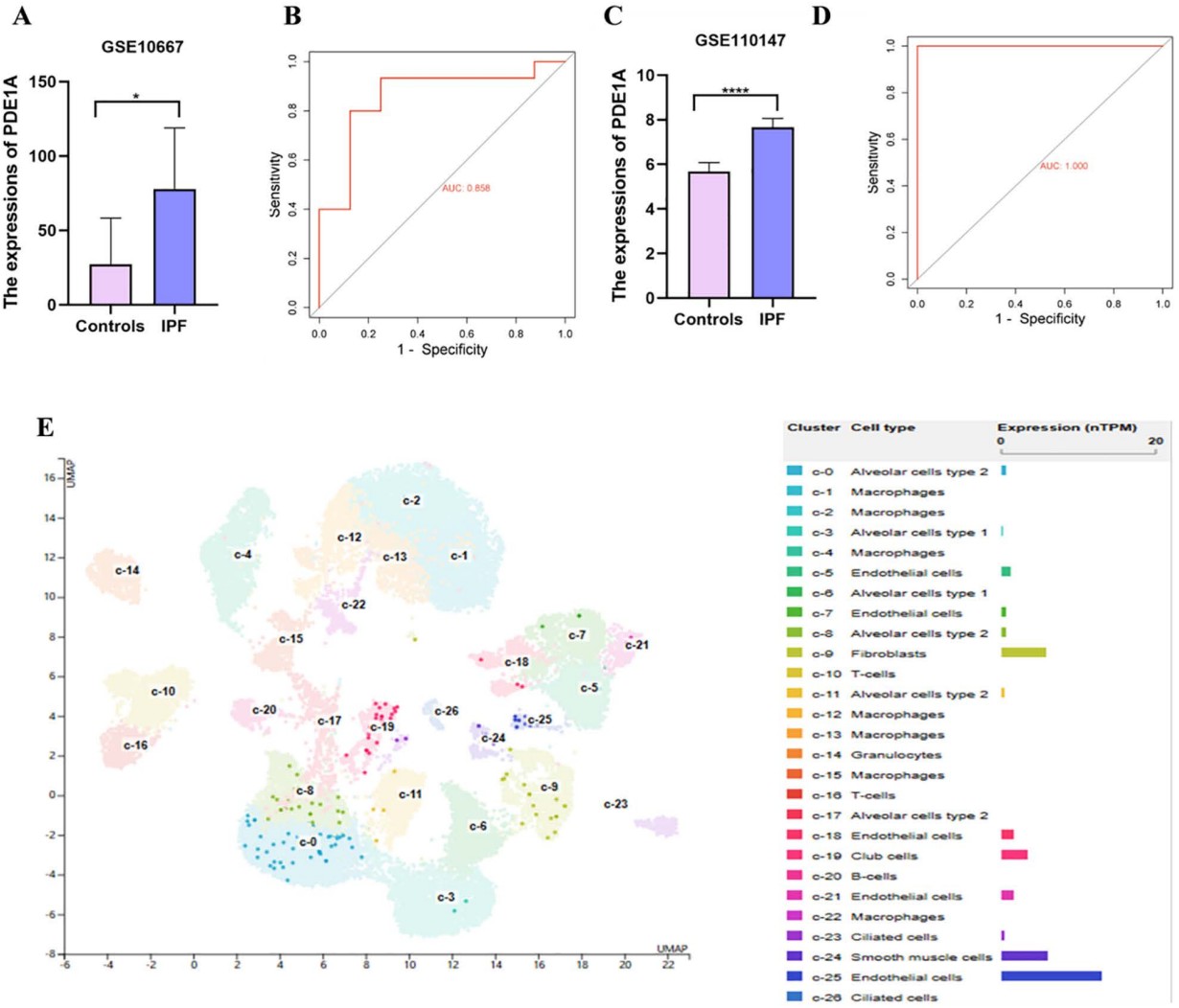

**Fig 4. Verification of core target genes.** A: The expression level of PDE1A in GSE10667 dataset; B: ROC curve of PDE1A in GSE10667 data set; C: Expression level of PDE1A in GSE110147 dataset; D: ROC curve of PDE1A in GSE110147 data set. E: The expression of PDE1A in different cells in the HAP database.

Overall, the RMSD of the complex stabilized earlier and fluctuated less than that of the protein monomer, indicating stable binding (Fig 6B). According to the RMSF analysis, amino acids at positions 260–269, 330–333, 366–367, and 384–388 showed fluctuations particularly in the loop region near the binding pocket. Throughout the 200 ns molecular dynamic simulation, the ligand remained stably positioned near the binding pocket indicating a stable complex (Fig 6C). Higher flexibility in several regions were observed through residue-wise RMSF analysis. Moreover, particularly 260–269, 330–333, 366–367, and 384–388 residues showed predominantly correspond to hinge and loop regions connecting secondary structure elements. These observed residues are structurally significant due to their role in conformational adaptability and also in the proximity of ligand-binding site. A notable reduction in RMSF values observed for the particular regions upon ligand binding through comparative analysis between pirfenidone-bound systems and the apo form. Reduced fluctuations in particular key regions revealed that the pirfenidone binding restricts the local protein dynamics leads to contribute to enhance the structural stability of PDE1A. The averaged values of RMSF were calculated for these regions and lower

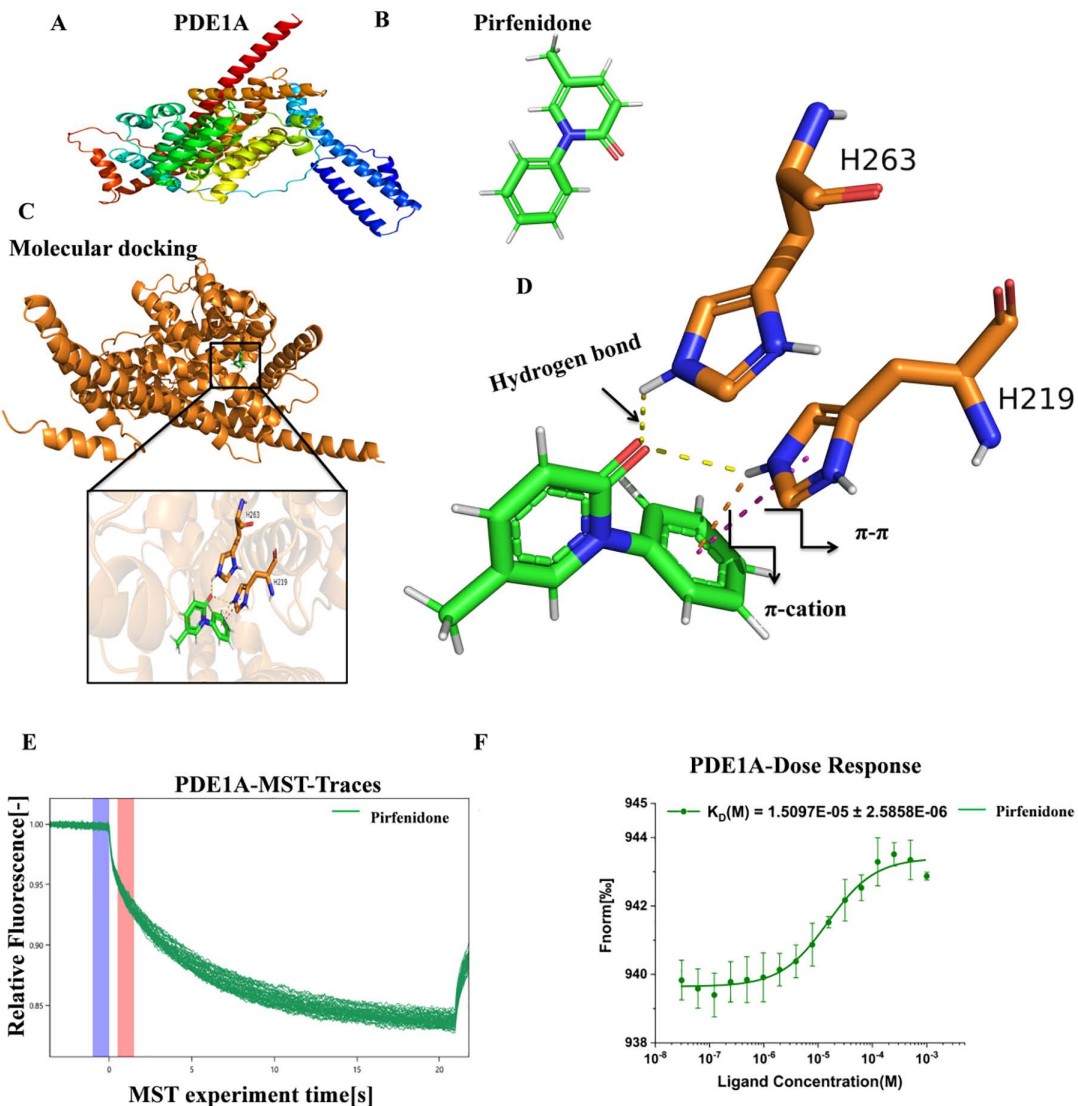

**Fig 5. The experimental results of MD and MST.** A: Structural diagram of protein PDE1A; B: Pirfenidone structure diagram; C: The docking structure diagram of Pirfenidone and PDE1A; D: 3D diagram of the interaction between Pirfenidone and PDE1A (affinity −6.4 kcal/mol), The green rod-like structure is a small molecule; The brown rod-like structure is the residue of the protein structure, Yellow dotted line: Hydrogen bond Pink dotted line: π-πinteraction; Orange dotted line: π-cation; E: Each curve in the figure is a real-time recording curve of fluorescence intensity in a capillary tube, which records the variation of fluorescence intensity over time in the temperature gradient. Purple represents the initial average signal F0, red represents the average signal F1 of the time period selected for the fitting graph, and the homogenized fluorescence signal Fnorm is the thousandth ratio of F1/F0. As the heating begins, the fluorescently labeled protein surges towards the surrounding low-temperature area. The density of the fluorescent protein per unit space decreases, so the Fnorm value will decrease. After the heating ends, the temperature recovery gradient disappears and the Fnorm value rebounds; F: Fnorm fitting graph.

fluctuations in the ligand-bound system compared to the apo form was observed, with reduced flexibility upon pirfenidone binding.

The radius of gyration (Rg) of the protein-ligand complex was lower than that of the monomer, suggesting increased compactness due to ligand binding (Fig 6D). A reduction in SASA further indicated that ligand binding buried exposed regions of the protein surface, emphasizing hydrophobic interactions in stabilizing the complex (Fig 6E).

**Table 1. Molecular docking analyses along with binding affinity.**

| Combination mode | Binding affinity (kcal/mo1) | RMSD (1ower bound) | RMSD (upper bound) |
|---|---|---|---|
| 1 | −6.4 | 0 | 0 |
| 2 | −6.3 | 3.025 | 3.737 |
| 3 | −6.3 | 2.388 | 5.651 |
| 4 | −6.3 | 2.444 | 5.564 |
| 5 | −6.3 | 2.818 | 4.441 |
| 6 | −6.3 | 3.439 | 4.844 |
| 7 | −6.2 | 2.063 | 5.665 |
| 8 | −6.1 | 1.781 | 2.105 |
| 9 | −6.0 | 2.100 | 3.173 |

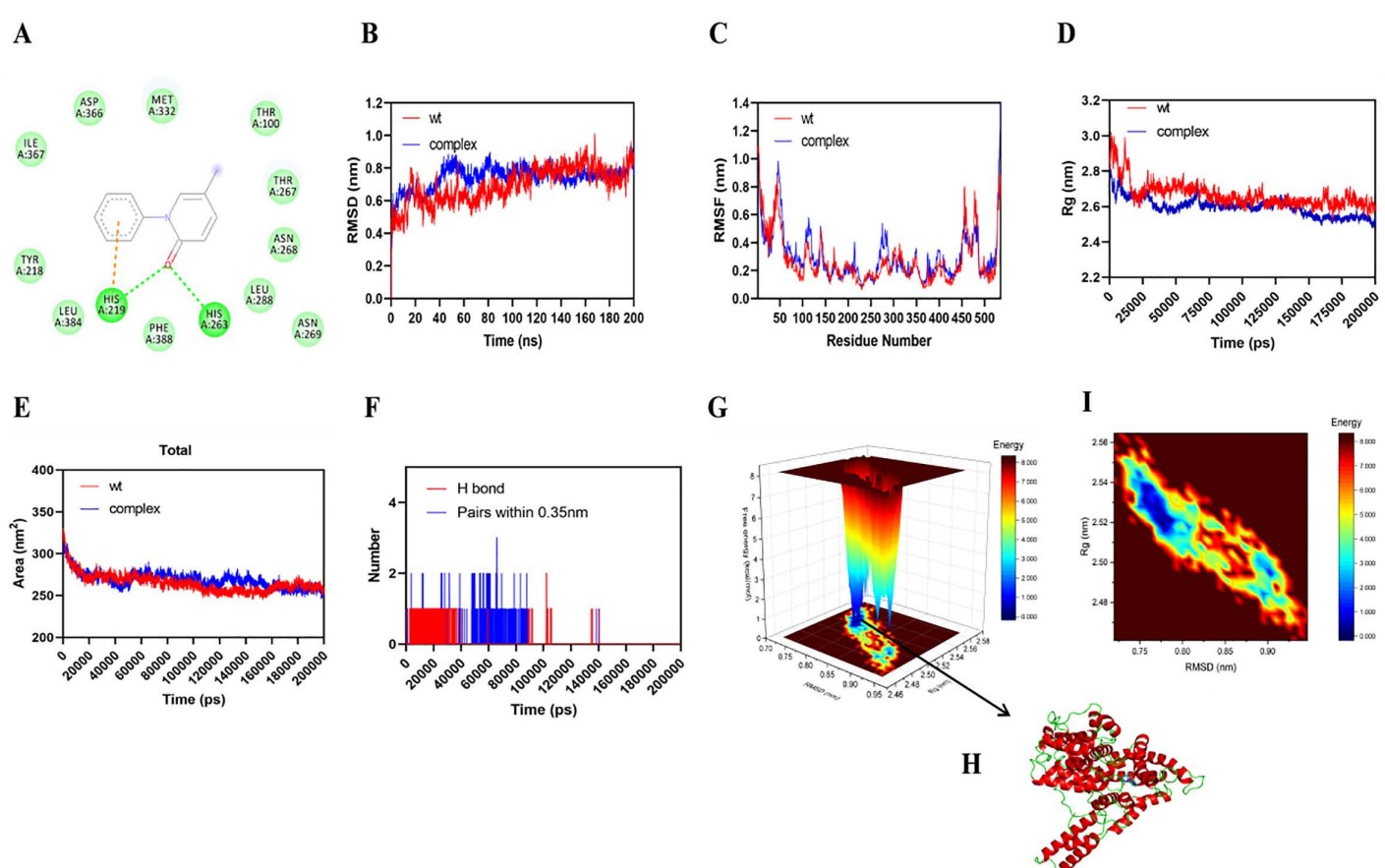

**Fig 6. The experimental results of MD and MST.** A: Proteins dock with small molecules; B: RMSD result graph; C: RMSF Fluctuation analysis; D: Circumferential radius result; E: Solvent accessible surface area; F: Number of hydrogen bonds; G-I: Free energy topography.

Hydrogen bond analysis showed that initial binding was suppoted by hydrogen bonds, which decreased significantly after 100 ns. This decrease may reflect ligand rearrangements, conformational shifts, or change in hydrogen bond geometry. A reduction in close contacts (pairs within 0.35 nm) supported this observation (Fig 6F). The free energy morphology

diagram indicated that the blue region corresponding to the lowest free energy state of the complex representing the the most stable conformation during the entire simulation (Fig 6G-6I).

Using gmx_MMPBSA, the average binding free energy of the complex was −14.67 kcal/mol (Table 2) supporting favourable binding.

### PDE1A expression in lung tissue and the impact of pirfenidone treatment

Validation through the GSE226249 dataset further confirmed the role of pirfenidone in modulating PDE1A expression (Fig 7A, 7C). Volcano plots comparing the normal vs. IPF groups and the IPF vs. IPF+ pirfenidone groups revealed significant changes in gene expression, highlighting the impact of pirfenidone on PDE1A regulation (Fig 7B, 7D). Comparative analysis demonstrated a significant increase in PDE1A expression in the IPF group compared with the normal group (Fig 7E). However, pirfenidone treatment resulted in a marked reduction in PDE1A levels, suggesting a therapeutic mechanism by which pirfenidone alleviates pulmonary fibrosis by downregulating PDE1A (Fig 7E). Taken together, these results suggest that pirfenidone may exert its therapeutic effect on IPF by specifically reducing PDE1A expression.

## Discussion

In this study, we used a comprehensive bioinformatics and experimental approach to investigate the molecular mechanisms by which pirfenidone exerts its therapeutic effect in IPF. Pirfenidone is an antifibrotic drug approved for the treatment of IPF, but its precise molecular targets are still not fully understood. By integrating diverse data sources and validation strategies, we sought to identify key targets and molecular interactions relevant to pirfenidone's mechanism of action. Using the GeneCards database, we identified 6,867 genes associated with IPF, reflecting the complex and multifactorial nature of the disease. At the same time, 101 potential pirfenidone targets were predicted using the SwissTargetPrediction tool. Cross-referencing these datasets revealed 71 overlapping genes, which we propose as important candidates for pirfenidone's therapeutic activity.

To investigate the functional relevance of these targets, we performed enrichment analysis using the Metascape platform. Overlapping genes were significantly enriched in biological processes including cyclic nucleotide and cAMP-mediated signaling pathways. Both play pivotal roles in regulating cell proliferation, differentiation, and fibrotic responses. These results are consistent with previous studies linking dysregulated cAMP signaling to the pathophysiology of fibrotic lung diseases [5]. Furthermore, KEGG pathway analysis revealed enrichment in calcium signaling, taste transduction, morphine dependence, purine metabolism, and renin secretion; some of these processes are associated with tissue remodeling and fibrosis [12].

To further refine our candidate list, we analyzed two publicly available microarray datasets (GSE10667 and GSE110147) for differentially expressed genes (DEGs) in IPF patients. Based on the overlap of the DEGs with IPF-associated genes and pirfenidone targets, we identified PDE1A as a hub gene of interest. PDE1A was significantly

**Table 2. Protein binding free energy with ligand (MM/PBSA analysis).**

| Energy component | Average |
| --- | --- |
| ΔVDWAALS | −25.62 Kcal/mol |
| ΔEEL | −13.92 Kcal/mol |
| ΔEPB | 27.71 Kcal/mol |
| ΔENPOLAR | −2.84 Kcal/mol |
| ΔGGAS | −39.54 Kcal/mol |
| ΔGSOLV | 24.87 Kcal/mol |
| ΔTOTAL | −14.67 Kcal/mol |

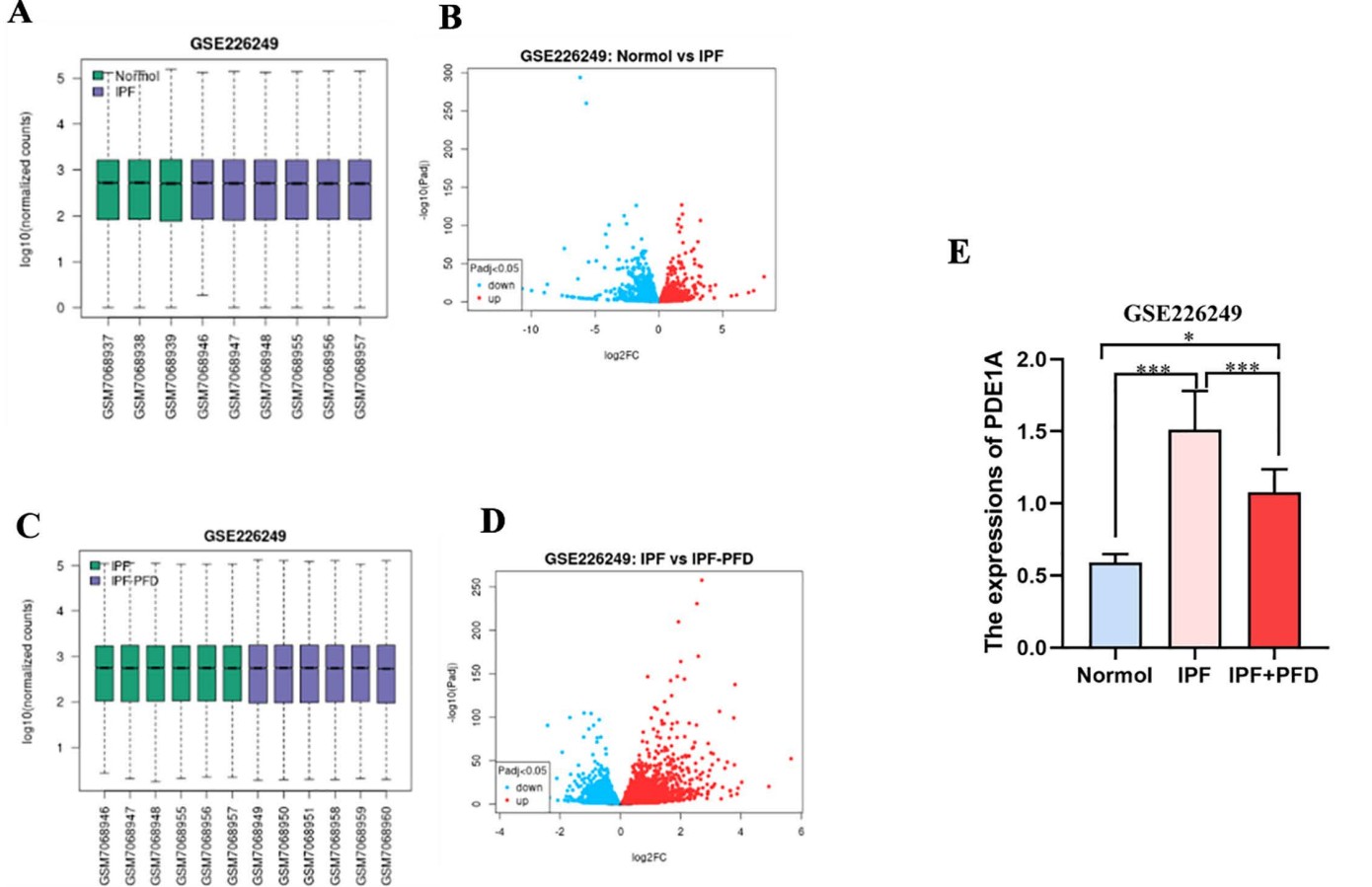

**Fig 7. Validation of PDE1A in the GSE226249 dataset.** A: box diagram of normal group and IPF group of GSE226249 data set; B; Volcano maps of normal group and IPF group in GSE226249 dataset; C: Box diagram of IPF and IPF+ pirfenidone group of GSE226249 data set; D; Volcanic maps of IPF and IPF+ pirfenidone groups in GSE226249 dataset; E: Expression of PDE1A in the GSE226249 dataset.

overexpressed in IPF lung tissue, and ROC analysis showed an area under the curve (AUC) greater than 0.8, suggesting high diagnostic potential. These results are in agreement with previous studies reporting increased PDE1A expression in fibrotic diseases, where it was linked to cyclic nucleotide degradation and the promotion of profibrotic signaling cascades [13].

Cell-specific expression analyses using the Human Protein Atlas (HPA) showed that PDE1A is predominantly expressed in lung fibroblasts, smooth muscle cells, and endothelial cells, the main cellular players in IPF. Fibroblasts, upon activation into myofibroblasts, produce excess extracellular matrix and thus directly contribute to the formation of fibrotic tissue. This conversion is known to be regulated by cytokines such as transforming growth factor-β (TGF-β) and platelet-derived growth factor (PDGF), both of which have previously been linked to PDE activity [14–16]. Endothelial cells, in turn, contribute to inflammation, angiogenesis, and the recruitment of immune cells, thus exacerbating fibrotic remodeling [17–20]. Although smooth muscle cells are not critical for the development of fibrosis, they contribute to vascular remodeling and can indirectly worsen tissue stiffness and vascular resistance [21].

To investigate the interaction between pirfenidone and PDE1A, we performed molecular docking using AutoDock Vina. The binding energy of −6.4 kcal/mol indicated a favorable interaction. Pirfenidone forms hydrogen bonds with amino acid

residues H219 and H263 of PDE1A and, through H219, also participates in π-cation and π-π stacking interactions. These non-covalent interactions suggest a specific and stable binding mode. Our results are supported by previous studies, such as Yang et al. (2023), which highlighted the role of PDE inhibitors in suppressing fibrotic signaling pathways in lung fibroblasts [22], although pirfenidone has not been previously validated as a specific PDE1A binder.

To validate these *in silico* results, we used microthermophoresis (MST) to experimentally determine the binding affinity between pirfenidone and PDE1A. The MST experiment yielded a signal-to-noise ratio (SNR) of 19.17, indicating high data quality. The resulting binding curve displayed a characteristic S-shape, confirming specific and saturable binding. The dissociation constant ($K_D$) calculated to be 15.1 µM, indicating moderate to strong affinity under physiological conditions. These results are consistent with docking predictions and support the proposed interaction. Furthermore, analysis of the GSE226249 dataset showed that pirfenidone treatment was associated with a reduction in PDE1A expression in IPF models. This suggests that the compound may exert its antifibrotic effects partly through PDE1A downregulation. This is consistent with previous pharmacological studies showing that pirfenidone inhibits TGF-β signaling and modulates downstream fibrotic pathways [23,24].

Nevertheless, this study efficiently uncovers potential binding sites and key amino acid residues. However, we acknowledge that this computational approach has certain limitations. First, AutoDock treats proteins primarily as rigid bodies during molecular docking, allowing only the flexible rotational degrees of freedom of ligands. This limitation can make it difficult to accurately capture true binding conformations, especially for protein targets with obvious conformation-inducing effects. Furthermore, the AutoDock scoring function is primarily based on empirical energy terms, including simplified energy calculation models such as van der Waals forces, electrostatic forces, hydrogen bonding, and hydrophobic interactions. While these energy terms are valuable for rapidly screening large numbers of ligands, the scoring function cannot comprehensively account for complex factors such as the effect of solvent on proteins and ligands in aqueous environments, conformational entropy changes, and adjustments in protein flexibility. Therefore, the predicted binding free energy often contains significant errors and does not accurately reflect the actual binding affinity. Using molecular docking technology, we computationally identified potential interactions between PDE1A and pirfenidone. We subsequently performed an MST experiment and determined a $K_D$ value of 15.1 µM for the interaction between PDE1A and pirfenidone. This value was in agreement with the molecular docking results and confirmed the interaction. Further studies, including mutagenesis, structural biology, and in vivo validation, are required to confirm the therapeutic relevance of PDE1A inhibition by pirfenidone.

## Conclusion

These findings not only provide a solid experimental basis for the application of pirfenidone in the treatment of IPF but also provide new clues for understanding its mechanism of action. Despite the progress made in this study, there are several limitations and shortcomings: the current study focused mainly on in vitro and computational analyses, and there is a lack of sufficient in vivo experimental data to directly prove the actual effect of pirfenidone through PDE1A in IPF treatment. Although this study revealed that PDE1A may be a target of pirfenidone, the detailed mechanism by which pirfenidone affects downstream signaling pathways and cell function through PDE1A has not been fully elucidated. Overall, these findings not only deepen our understanding of the mechanism of action of pirfenidone but also open new avenues for precision medicine in IPF. Future studies are needed to validate our findings through additional clinical trials and further explore the optimal application strategy of pirfenidone in the treatment of IPF.

## Author contributions

**Conceptualization:** Jing Wu, Xue Yi.

**Data curation:** Haseeb Khaliq, Sheikh Arslan Sehgal.

**Funding acquisition:** Xue Yi.

**Investigation:** Jing Wu.

**Methodology:** Jing Wu.

**Project administration:** Haseeb Khaliq, Xue Yi.

**Resources:** Yanyan Ke.

**Software:** Sheikh Arslan Sehgal.

**Validation:** Yanyan Ke.

**Visualization:** Yanyan Ke.

**Writing – original draft:** Yanyan Ke, Xue Yi.

**Writing – review & editing:** Haseeb Khaliq, Qudrat Ullah.

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
