## [Decision Letter · Decision Letter 0]

12 May 2025

Dear Dr. Ullah,

Thank you for submitting your manuscript to PLOS ONE. After careful consideration, we feel that it has merit but does not fully meet PLOS ONE’s publication criteria as it currently stands. Therefore, we invite you to submit a revised version of the manuscript that addresses the points raised during the review process.

We look forward to receiving your revised manuscript.

Kind regards,

Mahbub Hasan, PhD

Academic Editor

PLOS ONE

Journal Requirements:

Reviewers' comments:

Reviewer's Responses to Questions

**Comments to the Author**

1. Is the manuscript technically sound, and do the data support the conclusions?

Reviewer #1: No

Reviewer #2: Yes

2. Has the statistical analysis been performed appropriately and rigorously?

Reviewer #1: No

Reviewer #2: Yes

3. Have the authors made all data underlying the findings in their manuscript fully available?

Reviewer #1: No

Reviewer #2: Yes

4. Is the manuscript presented in an intelligible fashion and written in standard English?

Reviewer #1: No

Reviewer #2: No

Reviewer #1: After careful consideration of the manuscript titled "Integrated Network Pharmacology and Bioinformatics to identify a novel strategy for Pirfenidone targeting phosphodiesterase 1A in the Treatment of Idiopathic Pulmonary Fibrosis," I regret to inform that the work does not meet the scientific rigor required for publication in PLOS ONE.

I find the following Major Concerns

1.The In silico analysis presented in this manuscript falls short of established standards. While computational approaches are valuable, they require proper validation and detailed methodology to support conclusions.

2.The authors state they downloaded structures of seven differentially expressed genes from the RCSB Protein Data Bank but fail to provide the specific PDB IDs. This omission prevents proper assessment and reproducibility of the work.

3.The manuscript claims that a docking binding energy of less than −5 kcal/mol indicates strong binding affinity. This threshold is an oversimplification of molecular docking. Proper docking studies typically include comprehensive analysis of binding poses and conformations. The assertion that -5 kcal/mol represents strong binding is arbitrary without contextual validation.

4.No positive control was included in the docking experiments. For reliable docking protocols, re-docking of co-crystal ligands (whenever possible) with RMSD calculations below 2Å is typically used to validate methodology.

5.The manuscript lacks critical details regarding amino acid residues involved in the interaction with the ligand and the forms of interactions. A thorough docking analysis should include both bonding and non-bonding interactions and an examination of key contact points between the ligand and receptor.

6.The authors should perform in vitro validation of the binding or conduct molecular dynamics simulations of at least 200ns to assess the stability of the protein-ligand complex over time.

7.Discuss the limitations of computational approaches. The study relies on limited computational evidence without sufficient validation or detailed analysis.

Reviewer #2: Molecular docking was not done correctly!

1. structures of seven differentially expressed genes were obtained from the RCSB Protein Data Bank. Please mention the PDB ids) (PDB, https://www.rcsb.org/) and99

saved in PDB format.

2. CB-Dock2 conducts blind docking. Why not active site specific docking?

3. Before docking the co-crystal ligand should be redocked and rmsd with diagram should be mentioned in the supplementary material.

4. There are some differances between freeing binding energy and binding affinity. Please correct as because at the beginning "Molecular Docking" binding affinity was mentioned!

5. How docking binding energy of less than −5 kcal/mol was considered indicative of strong binding affinity?? is there any control drug used or how this threshold was decided>

6. I suggest to conduct mm-PBSA as because docking based mm-GBSA is not accurate to calculate free binding energy!

**Do you want your identity to be public for this peer review?** For information about this choice, including consent withdrawal, please see our For information about this choice, including consent withdrawal, please see our Privacy Policy .

Reviewer #1: No

Reviewer #2: No

While revising your submission, please upload your figure files to the Preflight Analysis and Conversion Engine (PACE) digital diagnostic tool, https://pacev2.apexcovantage.com/ . PACE helps ensure that figures meet PLOS requirements. To use PACE, you must first register as a user. Registration is free. Then, login and navigate to the UPLOAD tab, where you will find detailed instructions on how to use the tool. If you encounter any issues or have any questions when using PACE, please email PLOS at . PACE helps ensure that figures meet PLOS requirements. To use PACE, you must first register as a user. Registration is free. Then, login and navigate to the UPLOAD tab, where you will find detailed instructions on how to use the tool. If you encounter any issues or have any questions when using PACE, please email PLOS at figures@plos.org . Please note that Supporting Information files do not need this step.. Please note that Supporting Information files do not need this step.

---

## [Author Response · Author response to Decision Letter 1]

3 Jul 2025

Dear Editor,

We are extremely grateful to the editors and reviewers for the opportunity they have given us to revise the manuscript. We believe that through these supplements and revisions, our research can better meet the publication standards of PLOS ONE. We look forward to your further feedback and hope that the revised manuscript can be accepted for publication.

Response to Reviewer’s Comment:

Reviewer #1: Changes made according to reviewer 1 are highlighted in blue in the article.

1. The In-silico analysis presented in this manuscript falls short of established standards. While computational approaches are valuable, they require proper validation and detailed methodology to support conclusions.

Reply: Thank you for your valuable comments on our manuscript. We attach great importance to the issue you pointed out regarding the need for more rigorous validation and detailed methodological support for In silico analysis. We are fully aware of the deficiencies in the methodological description and result verification of the computer simulation part in the original manuscript. For this reason, in the revised draft:

(1) Clarify the technical details of molecular docking, including the acquisition of protein structures, determination of binding sites, and other technical details.

(2) Add experimental verification (MST). To enhance the reliability of the calculation results, we have newly added the micro thermal Migration (MST) experiment to verify the binding relationship between PDE1A and candidate compounds. The experimental results are consistent with the computational predictions, supporting the conclusion of molecular docking at the experimental level.

(3) In the molecular docking results, the analysis of the interaction between proteins and compounds has been added and the interaction has been visualized.

These results suggest that Pirfenidone may exert its biological effects by binding specifically to PDE1A. We have described in detail in the text the parameter Settings of molecular docking, the scoring system and the analysis of binding sites to ensure the transparency and repeatability of this part of the research. The modifications in the article are as follows: Interaction analysis of Pirfenidone and PDE1A: MD and MST

The binding patterns of PDE1A (Fig. 5A) and pirfenidone (Fig. 3B) were predicted through molecular docking (Fig. 5C). The docking results indicated that the binding energy between PDE1A and pirfenidone was -6.4 kcal/mol (Table 1), and they formed extensive interactions. Specifically, the side chains of amino acids H219 and H263 of PDE1A formed hydrogen bonds with pirfenidone. In addition, ligands also have π-π stacking and π- cation interactions with H219 residues. These bonding and non-bonding interactions work in synergy, enhancing the binding stability of the complex (Fig. 5D). In this study, we conducted a detailed analysis of the interaction between Pirfenidone and PDE1A through Microscale Thermophoresis (MST) technology. The experimental conditions are as follows: Ligand Concentration: 1000 µM to 0.0305 µM; Excitation Power: 100%; MST Power: 40%; Temperature: 25.0°C; Kd: 1.5097E-05; Kd Confidence: ± 2.5858E-06; Signal to Noise: 19.171163. The MST results show that a signal-to-noise ratio greater than 5 indicates a cutoff, while a ratio greater than 12 indicates high-quality data. Moreover, the protein mobility slows down with increasing analyte concentration. The fitting curve is S-shaped, exhibiting clear upper and lower plateau phases and strong concentration dependence; this indicates specific binding (Fig. 5E-F). The dissociation constant (Kd) is 15.1 μM, indicating relatively strong binding affinity in the in vitro molecular interaction between the protein and small molecules.

Tanble 1 Molecular docking scoring item

Combination mode Affinity (kcal/mo1) RMSD(1.b.) RMSD(u.b.)

1 -6.4 0 0

2 -6.3 3.025 3.737

3 -6.3 2.388 5.651

4 -6.3 2.444 5.564

5 -6.3 2.818 4.441

6 -6.3 3.439 4.844

7 -6.2 2.063 5.665

8 -6.1 1.781 2.105

9 -6.0 2.100 3.173

Fig. 5 The experimental results of MD and MST. A: Structural diagram of protein PDE1A; B: Pirfenidone structure diagram; C: The docking structure diagram of Pirfenidone and PDE1A; D: 3D diagram of the interaction between Pirfenidone and PDE1A (affinity -6.4 kcal/mol), The green rod-like structure is a small molecule; The brown rod-like structure is the residue of the protein structure, Yellow dotted line: Hydrogen bond Pink dotted line: π-πinteraction; Orange dotted line: π-cation; E: Each curve in the figure is a real-time recording curve of fluorescence intensity in a capillary tube, which records the variation of fluorescence intensity over time in the temperature gradient. Purple represents the initial average signal F0, red represents the average signal F1 of the time period selected for the fitting graph, and the homogenized fluorescence signal Fnorm is the thousandth ratio of F1/F0. As the heating begins, the fluorescently labeled protein surges towards the surrounding low-temperature area. The density of the fluorescent protein per unit space decreases, so the Fnorm value will decrease. After the heating ends, the temperature recovery gradient disappears and the Fnorm value rebounds; F: Fnorm fitting graph.

The methods in the article have been re-supplemented:

Molecular Docking (MD)

Macromolecular docking is one of the important methods of molecular simulation. It is a type of computer simulation. Its essence is the recognition process between two or more molecules. This process involves shape and spatial complementarity as well as energy compatibility between molecules. In the field of molecular modeling, if two molecules can form a stable complex, molecular docking can predict their binding mode. Furthermore, the binding strength can be evaluated using a scoring function based on this mode.

The software adopted this time is AutoDock Vina. The algorithm of AutoDock Vina is the Monte Carlo search algorithm based on local perturbation. The method of ligand docking with PDE1A (phosphodiesterase 1A, an enzyme involved in cellular signaling) is semi-flexible docking, that is, the ligand is flexible (the rotatable bond angle can rotate freely), while the PDE1A protein is rigid (remains unchanged). The 3D structure file of the receptor protein was downloaded from the PDB, and the protein structure was completed and optimized. If there is no crystal structure of the experimental data, AlphaFold3 is used to predict the 3D structure of the receptor protein (PDE1A: UniProt ID P5475). Download the 3D structure file of the small molecule compound from the PubChem database and perform structure optimization on the compound (Pirfenidone: PubChem ID 40632). If the structure file is unavailable, the StoneMIND Collector is used to draw the compound, generate the 3D structure, and then perform energy minimization. The active pockets of proteins are determined by integrating information provided by users, annotations from the UniProt database, and AI predictions. This project uses KVFinder[21] to predict the active pockets of proteins, determines the active sites of the proteins, and performs molecular docking using AutoDock Vina. After processing and optimizing the docking data, the results were analyzed using Discovery Studio Visualizer.

MicroScale Thermophoresis (MST)

MicroScale Thermophoresis (MST) is a powerful technique for quantifying interactions between biomolecules. MST combines precise fluorescence detection with sensitive and flexible thermophoresis to provide a powerful and rapid method for measuring intermolecular interactions. Specifically, during the MST experiment, the sample is heated by an infrared laser to generate a microscopic temperature gradient. Then, the directional movement of molecules is monitored and quantified through covalently bound fluorescent dyes or the autofluorescence of tryptophan. The application scope includes interactions involving small molecules, proteins, and protein complexes.

(1) Protein labeling (NTA dye)

Take a new clean 1.5 mL centrifuge tube, add 2 µL of the newly prepared RED-tris-NTA second-generation dye and 98 µL of PBS-T, and gently pipette and mix evenly with a pipette to obtain a dye solution with a final concentration of 100 nM. Then add 100 µL of 200nM protein sample. After wrapping the centrifuge tubes with tin foil, incubate them at room temperature in the dark for 30 minutes in the dark.

After the protein labeling step is completed, centrifuge at 4 ° C and 15,000 g for 10 minutes, and take the supernatant into a new centrifuge tube.

(2) Protein labeling fluorescence intensity detection (Pretest)

After fluorescence labeling, 200 μL of protein was obtained. 10μL was taken out and diluted 3 times with Assay Buffer. Open the sample chamber, take out the tray, draw the diluted protein sample with a capillary tube and place it in the corresponding slot of tray 1.2. Input protein sample information and Assay Buffer information; Select the capillary type; Excitation Power: Select Auto; MST-Power selects Medium.

(3) Binding Affinity determination

Dilute the analyte to twice the maximum final loading concentration of 100μL using the Assay Buffer. Take two eight-row samples, make labels 1-16, and add 10μL of Assay Buffer to tubes 2-16 respectively; Add 20μL of the prepared analyte to tube 1, and use a pipette to draw 10 μL. Start the 2-fold gradient dilution successively in tubes 2 to 16. Then, starting with a pipette, add 10μL of protein to each tube and pipette to mix well. After the mixed sample is incubated at room temperature for 15 minutes, it is aspirated with a capillary tube and loaded onto the sample tray. Then, click "Start" on the software to begin the experiment.

Data analysis was conducted after repeating the steps of dilution - protein mixing - sample loading for a total of three times.

2. The authors state they downloaded structures of seven differentially expressed genes from the RCSB Protein Data Bank but fail to provide the specific PDB IDs. This omission prevents proper assessment and reproducibility of the work.

Reply: Thank you very much for your attention and valuable suggestions to our manuscript. When we were writing the method section, the information was incorrect due to a typo. This is entirely due to our negligence. Thank you for pointing out this problem. We have made modifications in the method section and supplemented the specific PDB IDs. The following is the revised description: The software adopted this time is AutoDock Vina. The algorithm of AutoDock Vina is the Monte Carlo search algorithm based on local perturbation. The method of ligand docking with PDE1A (phosphodiesterase 1A, an enzyme involved in cellular signaling) is semi-flexible docking, that is, the ligand is flexible (the rotatable bond angle can rotate freely), while the PDE1A protein is rigid (remains unchanged). The 3D structure file of the receptor protein was downloaded from the PDB, and the protein structure was completed and optimized. If there is no crystal structure of the experimental data, AlphaFold3 is used to predict the 3D structure of the receptor protein (PDE1A: UniProt ID P5475). Download the 3D structure file of the small molecule compound from the PubChem database and perform structure optimization on the compound (Pirfenidone: PubChem ID 40632).

3. The manuscript claims that a docking binding energy of less than −5 kcal/mol indicates strong binding affinity. This threshold is an oversimplification of molecular docking. Proper docking studies typically include comprehensive analysis of binding poses and conformations. The assertion that -5 kcal/mol represents strong binding is arbitrary without contextual validation.

Reply: Thank you very much for your attention and valuable suggestions to our manuscript. We have deleted the expression "A binding energy less than -5 kcal/mol is considered strong binding" in the original text and changed it to a more cautious description, such as: "Generally, a binding energy less than -5 kcal/mol can be regarded as having certain binding potential, but the specific binding ability still needs to be verified in combination with other experiments." We supplemented the multi-angle analysis of the docking results, including specific interactions such as hydrogen bonds, hydrophobic interactions, and π-π packing, and combined with visual images to present the binding mode between ligands and targets. The modifications in the article are as follows:

Interaction analysis of Pirfenidone and PDE1A: MD and MST

The binding patterns of PDE1A (Fig. 5A) and pirfenidone (Fig. 3B) were predicted through molecular docking (Fig. 5C). The docking results indicated that the binding energy between PDE1A and pirfenidone was -6.4 kcal/mol (Table 1), and they formed extensive interactions. Specifically, the side chains of amino acids H219 and H263 of PDE1A formed hydrogen bonds with pirfenidone. In addition, ligands also have π-π stacking and π- cation interactions with H219 residues. These bonding and non-bonding interactions work in synergy, enhancing the binding stability of the complex (Fig. 5D). In this study, we conducted a detailed analysis of the interaction between Pirfenidone and PDE1A through Microscale Thermophoresis (MST) technology. The experimental conditions are as follows: Ligand Concentration: 1000 µM to 0.0305 µM; Excitation Power: 100%; MST Power: 40%; Temperature: 25.0°C; Kd: 1.5097E-05; Kd Confidence: ± 2.5858E-06; Signal to Noise: 19.171163. The MST results show that a signal-to-noise ratio greater than 5 indicates a cutoff, while a ratio greater than 12 indicates high-quality data. Moreover, the protein mobility slows down with increasing analyte concentration. The fitting curve is S-shaped, exhibiting clear upper and lower plateau phases and strong concentration dependence; this indicates specific binding (Fig. 5E-F). The dissociation constant (Kd) is 15.1 μM, indicating relatively strong binding affinity in the in vitro molecular interaction between the protein and small molecules.

4. No positive control was included in the docking experiments. For reliable docking protocols, re-docking of co-crystal ligands (whenever possible) with RMSD calculations below 2Å is typically used to validate methodology.

Reply: Thank you very much for your attention and valuable comments on our manuscript. However, due to the lack of co-crystalline data of PDE1A protein, re-docking cannot be carried out. In order to improve the accuracy of the molecular docking method, we have made the following adjustments.

At present, there is no relevant co-crystalline structure data for PDE1A (UniProt number: P54750), nor is there experimental crystal structure information of the protein. Meanwhile, there is no recognized positive control drug yet. Therefore, re-docking verification based on the binding sites of positive compounds cannot be carried out.

2. In this study, AlphaFold3 was used to predict the three-dimensional structure of PDE1A (with a pTM score of 0.75), and KVFinder was utilized to predict potential compound binding sites. Based on this binding site, molecular docking was carried out using AutoDock Vina. Subsequently, the interaction between the protein and the ligand was analyzed in detail with the help of Discovery Studio Client, and the visualization display of the binding mode was achieved by using PyMOL.

5. The manuscript lacks critical details regarding amino acid residues involved in the interaction with the ligand and the forms of interactions. A thorough docking analysis should include both bonding and non-bonding interactions and an examination of key contact points between the ligand and receptor.

Reply: Thank you to the reviewers for your valuable comments. We supplemented the detailed analysis of the molecular docking results and further clarified the key interactions between ligands and receptor proteins. Calculated by AutoDock Vina, the binding free energy of PDE1A to the compound was -6.4 kcal/mol, showing a good binding potential. The specific binding mode analysis indicates that the ligand has formed stable hydrogen bond interactions with the key amino acid residues H219 and H263 of PDE1A. In addition, ligands also have π-π stacking

---

## [Decision Letter · Decision Letter 1]

30 Jul 2025

Dear Dr. Ullah,

**Please revise the manuscript as reviewer 2's suggestions.** Please submit your revised manuscript by Sep 12 2025 11:59PM. If you will need more time than this to complete your revisions, please reply to this message or contact the journal office at plosone@plos.org . . A rebuttal letter that responds to each point raised by the academic editor and reviewer(s). You should upload this letter as a separate file labeled 'Response to Reviewers'.A marked-up copy of your manuscript that highlights changes made to the original version. You should upload this as a separate file labeled 'Revised Manuscript with Track Changes'.An unmarked version of your revised paper without tracked changes. You should upload this as a separate file labeled 'Manuscript'.

We look forward to receiving your revised manuscript.

Kind regards,

Mahbub Hasan, PhD

Academic Editor

PLOS ONE

**Journal Requirements:**

Reviewers' comments:

Reviewer's Responses to Questions

**Comments to the Author**

Reviewer #1: All comments have been addressed

Reviewer #2: All comments have been addressed

2. Is the manuscript technically sound, and do the data support the conclusions?

Reviewer #1: Yes

Reviewer #2: Partly

3. Has the statistical analysis been performed appropriately and rigorously?

Reviewer #1: Yes

Reviewer #2: N/A

4. Have the authors made all data underlying the findings in their manuscript fully available?

Reviewer #1: Yes

Reviewer #2: Yes

5. Is the manuscript presented in an intelligible fashion and written in standard English?

Reviewer #1: Yes

Reviewer #2: Yes

**Reviewer #1:**  Dear author, Dear author,

The manuscript has been improved, and all my recommendations have been addressed. I recommend the manuscript be accepted in its current form.

**Reviewer #2:**  Revision is not satisfactory. Revision is not satisfactory.

1. Regarding binding energy reviewer 1 also highlighted the same. I suggest that the authors may take 3 months time use good software such as Autodock Vina, SwissDock or any other molecular docking software and perform the docking correctly.

2. As mm-gbsa did not provided good results, question is arising about the authenticity of the data. I suggest to conduct "Consensus molecular docking" using multiple docking software and identify the best pose.

3. Take an extension of 3-4 months and try to conduct 200 to 500 ns Molecular dynamic simulation to understand the Protein-Ligand interaction dynamics.

4. I have a doubt whether the author chosen the right protein target or not! Authors should also think about it and if found true then change it accordingly. With current results it is difficult to agree with the conclusion.

5. Reply to point 5 and 6 is same, whereas the context of the points were different!

**Do you want your identity to be public for this peer review?** For information about this choice, including consent withdrawal, please see our For information about this choice, including consent withdrawal, please see our Privacy Policy .

Reviewer #1: No

Reviewer #2: No

While revising your submission, please upload your figure files to the Preflight Analysis and Conversion Engine (PACE) digital diagnostic tool, https://pacev2.apexcovantage.com/ . PACE helps ensure that figures meet PLOS requirements. To use PACE, you must first register as a user. Registration is free. Then, login and navigate to the UPLOAD tab, where you will find detailed instructions on how to use the tool. If you encounter any issues or have any questions when using PACE, please email PLOS at . PACE helps ensure that figures meet PLOS requirements. To use PACE, you must first register as a user. Registration is free. Then, login and navigate to the UPLOAD tab, where you will find detailed instructions on how to use the tool. If you encounter any issues or have any questions when using PACE, please email PLOS at figures@plos.org . Please note that Supporting Information files do not need this step.. Please note that Supporting Information files do not need this step.

---

## [Author Response · Author response to Decision Letter 2]

6 Sep 2025

Dear editor,

A proper Response to Reviewer's comments has been provided as attached file.

Reviewer #1

1. Regarding binding energy reviewer 1 also highlighted the same. I suggest that the authors may take 3 months’ time use good software such as Autodock Vina, SwissDock or any other molecular docking software and perform the docking correctly.

Reply: Thank you for your valuable comments on our manuscript.

We used software such as Discovery studio2019, Schrodinger2021, and Swiss-dock to re-perform molecular docking. The connection result is as follows:

Software Binding energy

Autodock vina -6.4 kcal/mol

Discovery studio2019 -8.124 kcal/mol

Schrodinger2021 -4.41349 kcal/mol

Swiss-dock -6.2662 kcal/mol

This is the best conformation given by different software:

2. As mm-gbsa did not provided good results, question is arising about the authenticity of the data. I suggest to conduct "Consensus molecular docking" using multiple docking software and identify the best pose.

Reply: Thank you very much for your attention and valuable suggestions to our manuscript. We have answered this question in Question 1

3. Take an extension of 3-4 months and try to conduct 200 to 500 ns Molecular dynamic simulation to understand the Protein-Ligand interaction dynamics.

Reply: Thank you very much for your attention and valuable suggestions to our manuscript. We conducted a new molecular dynamics simulation. The simulation video is available on PDE1A.mp4 and has been modified as follows in the article:

We conducted the analysis using Discovery Studio Visualizer: Through the analysis of the docking results, hydrogen bond interactions were formed between his219 and HIS263 of the PDE1A protein and the small molecule, while HIS219 and the small molecule formed Pi-Pi and Pi-cation interactions. The van der Waals forces provided by the amino acids around the small molecule jointly stabilized the small molecule in the binding pocket (Figure 6A). The first frame of the dynamic simulation trajectory was selected as a reference to calculate the root mean square deviation (RMSD) of Cα atoms. In the initial stage (0-20 ns), the RMSD of both rapidly increased, indicating that the system had adjusted from the initial conformation to the equilibrium state. Subsequently, the RMSD fluctuation range of the complex was mainly around 0.6-0.8 nm, showing a relatively stable trend, although there were still small fluctuations. The RMSD fluctuation range of the protein monomer is slightly wider, approximately 0.6-1.0 nm, and it shows an upward trend in the later stage of the simulation (160-200 ns). Overall, the RMSD of the complex seems to stabilize earlier than that of the protein monomers, and the fluctuation range is also smaller. The overall fluctuations of the protein monomers and complexes were within the acceptable range, indicating stable binding (Figure 6B). According to the RMSF comparison between wt and complex (amino acids 5A away from the ligand molecule), among the amino acids at positions 260-269, 330-333, 366-367, and 384-388 that constitute the ligand-binding pocket, the loop region composed of 260-269 and 330-333 fluctuates. In the protein-ligand complex, the ligand remained stable near the binding pocket throughout the 200ns kinetic simulation, indicating that the ligand formed a stable complex structure with the protein under these simulation conditions (Figure 6C). In this study, the Rg of the protein-ligand complex was lower than that of the protein monomer, reflecting that the persistent presence of the ligand in the binding pocket enhanced the compactness of the protein through local structural optimization and non-polar interactions (Figure 6D). In this study, the reduction of the complex SASA indicates that ligand binding buried part of the protein surface, reducing solvent accessibility. The stability of the pockets further confirmed that this change was caused by the ligand occupying the exposed area, highlighting the crucial role of hydrophobic interactions in maintaining the stability of the complex (Figure 6E). In this study, the charts show that the initial binding may rely on hydrogen bonds, but the number of hydrogen bonds decreased significantly after 100 ns. The disappearance of hydrogen bonds may be related to the dissociation of ligands at binding sites, conformational changes, or the rearrangement of hydrogen bond donors/acceptors. The reduction of Pairs within 0.35 nm further supports the reduction of close contact between ligands and proteins. The bifurcation of the red line (H-bond) and the blue line (Pairs within 0.35 nm) in the figure indicates that some pairs may still exist but no longer meet the geometric requirements of hydrogen bonds (Figure 6F). As shown in the figure, the free energy morphology diagram was drawn with the RMSD and Rg trajectories at the last 10ns of the 200ns simulation. In the 2D diagram, the blue area indicates the lowest free energy point of protein-ligand binding, meaning that the protein-ligand complex at this site is the most stable conformation with the lowest free energy point throughout the simulation process (Figure 6G-I).

Fig. 6 The experimental results of MD and MST. A: Proteins dock with small molecules; B: RMSD result graph; C: RMSF Fluctuation analysis; D: Circumferential radius result; E: Solvent accessible surface area; F: Number of hydrogen bonds

; G-I: Free energy topography.

In this study, gmx_MMPBSA was used to calculate the free energy with Gromacs trajectories and topological files. The total binding energy of each complex was contributed by different components, including VDWAALS, EEL, EPB, ENPOLAR, GGAS and GSOLV. According to the results of RMSD, The trajectory of the last 10ns period after ligand stabilization was selected for MM/PBSA analysis. From the calculation results, it can be known that the average binding free energy of the two is -14.67 kcal/mol (Table 2).

Table 2 Protein binding free energy with small molecules (MM/PBSA analysis)

Energy Component Average

ΔVDWAALS -25.62

ΔEEL -13.92

ΔEPB 27.71

ΔENPOLAR -2.84

ΔGGAS -39.54

ΔGSOLV 24.87

ΔTOTAL -14.67

Corrections are added in page14, line 252.

4. I have a doubt whether the author chosen the right protein target or not! Authors should also think about it and if found true then change it accordingly. With current results it is difficult to agree with the conclusion.

Reply: Thank you very much for your attention and valuable comments on our manuscript. We have re-conducted the molecular dynamics experiment, and the results have been replied in question 3.

5. Reply to point 5 and 6 is same, whereas the context of the points were different!

Reply: Thank you to the reviewers for your valuable comments. We conducted molecular dynamics simulations of at least 200 nanoseconds to evaluate the temporal stability of protein-ligand complexes. The simulation results can be seen in the video file PDE1A.mp4.

---

## [Decision Letter · Decision Letter 2]

9 Oct 2025

Dear Dr. Ullah,

Thank you for submitting your manuscript to PLOS ONE. After careful consideration, we feel that it has merit but does not fully meet PLOS ONE’s publication criteria as it currently stands. Therefore, we invite you to submit a revised version of the manuscript that addresses the points raised during the review process.

We look forward to receiving your revised manuscript.

Kind regards,

Mahbub Hasan, PhD

Academic Editor

PLOS ONE

Journal Requirements:

Reviewers' comments:

Reviewer's Responses to Questions

**Comments to the Author**

Reviewer #2: All comments have been addressed

Reviewer #3: (No Response)

2. Is the manuscript technically sound, and do the data support the conclusions?

Reviewer #2: No

Reviewer #3: Partly

3. Has the statistical analysis been performed appropriately and rigorously?

Reviewer #2: No

Reviewer #3: No

4. Have the authors made all data underlying the findings in their manuscript fully available?

Reviewer #2: No

Reviewer #3: No

5. Is the manuscript presented in an intelligible fashion and written in standard English?

Reviewer #2: No

Reviewer #3: Yes

Reviewer #2: 1. MD Simulation procedure is not correct. Discovery Studio Visualizer is just a platform, NAMD available in Discovery Studio Visualizer can be used for MD Simulation.

2. Fig 6 F suggest after 140 ns there are no hydrogen bond interaction between Protein and ligand, which means the ligand is coming out from the binding pocket of the protein. This result undermine the conclusion.

3. Line 111 Molecular Docking (MD) is not correct. MD always refers to Molecular Dynamics Simulation.

4. In reply authors mentioned about Swiss-dock, but in whole paper authors did not mentioned anything about Swiss-dock or other software!

Reviewer #3: (No Response)

**Do you want your identity to be public for this peer review?** For information about this choice, including consent withdrawal, please see our For information about this choice, including consent withdrawal, please see our Privacy Policy .

Reviewer #2: No

Reviewer #3: No

While revising your submission, please upload your figure files to the Preflight Analysis and Conversion Engine (PACE) digital diagnostic tool, https://pacev2.apexcovantage.com/ . PACE helps ensure that figures meet PLOS requirements. To use PACE, you must first register as a user. Registration is free. Then, login and navigate to the UPLOAD tab, where you will find detailed instructions on how to use the tool. If you encounter any issues or have any questions when using PACE, please email PLOS at . PACE helps ensure that figures meet PLOS requirements. To use PACE, you must first register as a user. Registration is free. Then, login and navigate to the UPLOAD tab, where you will find detailed instructions on how to use the tool. If you encounter any issues or have any questions when using PACE, please email PLOS at figures@plos.org . Please note that Supporting Information files do not need this step.. Please note that Supporting Information files do not need this step.

---

## [Author Response · Author response to Decision Letter 3]

18 Dec 2025

We sincerely thank the reviewer for the constructive and detailed comments. All observations have significantly improved the quality, clarity, and reproducibility of our manuscript. We have carefully revised the manuscript as per suggestion and tried our best to incorporate all recommended changes.

Reviewer Comment:

The study is relevant but lacks reproducibility and methodological rigor. Additional methodological detail, MD validation, improved references, identifiers, and data/code availability are required.

Response: We have now substantially expanded the methodological descriptions, added detailed molecular dynamics (MD) analyses, corrected references, and included explicit dataset identifiers. All the generated data is available in the public repositories and the accession numbers are mentioned in the manuscript. These modifications have been fully incorporated into the revised manuscript.

Reviewer Comment: Provide step-by-step methodological detail.

Response: A comprehensive workflow description has been added, covering gene identification, target prediction, enrichment analysis, docking procedures, and validation steps. This has been incorporated throughout the Methods section.

DEG criteria

We have now specified thresholds, adjusted p-values, and the Benjamini–Hochberg method for multiple testing. This information is added.

GEO processing

Platform IDs and normalization approach covariates have been described in detail. Implemented in Methods.

Tools/versions

Exact software, versions, and packages (where applicable) used for analysis and data processing have been added to ensure reproducibility. Incorporated in Methods.

Justification for PDE1A selection

We expanded the explanation for prioritizing PDE1A among the seven DEGs based on functional importance, enrichment relevance, and reported links with fibrosis. Added where applicable.

Pose validation

Pose clustering method, key residues, and selection criteria have been added. Incorporated at appropriate position in the manuscript.

Functional Enrichment

We now specify ontology categories (GO BP/CC/MF), KEGG and Reactome usage, significance cutoffs, background gene set, and multiple-testing corrections. Included in manuscript.

Statistics

Sample sizes, test types, normality checks, and two-sided assumptions are now clearly stated. Incorporated in manuscript.

Reviewer Comment: Some references were not aligned and formatting needed correction.

Response: All references have been rechecked and reformatted.

Reviewer Comment: Docking alone is insufficient; MD simulations must be performed and reported.

Response: We fully agree. We have now performed molecular dynamics simulations using GROMACS and Ligand RMSD, Protein RMSF, Hydrogen-bond and hydrophobic interaction occupancies, Water-bridge analysis, MM/PBSA binding energies (mean ± SD) and other necessary analyses have been performed. All MD methodology and results have been integrated into the manuscript, including new figures and tables.

Regulatory approval

We have now specified the agency and year of pirfenidone approval (FDA 2014; EMA 2011) with primary citations.

MD integration

The MD findings are now integrated at appropriate positions in the manuscript.

We have added PDB IDs and UniProt IDs for PDE1A, GEO dataset IDs with GSM groupings, platform IDs, and Links of the utilized public repositories.

All identifiers are now clearly listed in the manuscript.

Response: In compliance with the reviewer’s request, we have explained the GEO preprocessing docking protocols with tool names, parameters and versions; analyze MD trajectories and mentioned the results in the manuscript at appropriate positions, topologies, and analysis code. DOI included in manuscript where needed. This ensures full transparency and reproducibility.

All requested analyses have been performed and added, including RMSD, RMSF, MM/PBSA binding energies, and other necessary analyses.

These results are presented in the revised Results.

We are grateful for the reviewer’s detailed and rigorous comments. All recommended revisions have been thoroughly implemented, and the manuscript has been significantly strengthened in terms of methodology, reproducibility, scientific depth, and clarity. We have incorporated each modification into the revised manuscript as requested.

---

## [Decision Letter · Decision Letter 3]

13 Jan 2026

Dear Dr. Ullah,

plosone@plos.org . . A letter that responds to each point raised by the academic editor and reviewer(s). You should upload this letter as a separate file labeled 'Response to Reviewers'.A marked-up copy of your manuscript that highlights changes made to the original version. You should upload this as a separate file labeled 'Revised Manuscript with Track Changes'.An unmarked version of your revised paper without tracked changes. You should upload this as a separate file labeled 'Manuscript'.

We look forward to receiving your revised manuscript.

Kind regards,

Mahbub Hasan, PhD

Academic Editor

PLOS One

Journal Requirements:

Reviewers' comments:

Reviewer's Responses to Questions

**Comments to the Author**

Reviewer #3: All comments have been addressed

2. Is the manuscript technically sound, and do the data support the conclusions?

Reviewer #3: Yes

3. Has the statistical analysis been performed appropriately and rigorously?

Reviewer #3: Yes

4. Have the authors made all data underlying the findings in their manuscript fully available?

Reviewer #3: No

5. Is the manuscript presented in an intelligible fashion and written in standard English?

Reviewer #3: No

Reviewer #3: analyses are generally satisfactory with the addition of simulation part, though all the mentioned recommendations were not done! However, as a minor point, the RMSF analysis could be further strengthened by a more residue-specific interpretation. In particular, the authors may consider explicitly highlighting and discussing the functional relevance of the residues showing elevated fluctuations (e.g., regions 260–269, 330–333, 366–367, and 384–388) in relation to secondary structure elements and proximity to the binding pocket. Additionally, a brief quantitative comparison of RMSF values between the apo and ligand-bound systems for these regions (e.g., ΔRMSF or averaged RMSF over the binding-site residues) would help clarify whether pirfenidone binding locally stabilizes PDE1A dynamics. These additions would improve clarity without requiring additional simulations.

**Do you want your identity to be public for this peer review?** For information about this choice, including consent withdrawal, please see our For information about this choice, including consent withdrawal, please see our Privacy Policy .

Reviewer #3: No

---

## [Author Response · Author response to Decision Letter 4]

21 Jan 2026

Response to Reviewer Comments

We sincerely thank the reviewer for the constructive and detailed comments. All observations have significantly improved the quality, clarity, and reproducibility of our manuscript. We have carefully revised the manuscript as per suggestion and tried our best to incorporate all recommended changes.

Reviewer Comment:

The study is relevant but lacks reproducibility and methodological rigor. Additional methodological detail, MD validation, improved references, identifiers, and data/code availability are required.

Response: We have now substantially expanded the methodological descriptions, added detailed molecular dynamics (MD) analyses, corrected references, and included explicit dataset identifiers. All the generated data is available in the public repositories and the accession numbers are mentioned in the manuscript. These modifications have been fully incorporated into the revised manuscript.

Reviewer Comment: Provide step-by-step methodological detail.

Response: A comprehensive workflow description has been added, covering gene identification, target prediction, enrichment analysis, docking procedures, and validation steps. This has been incorporated throughout the Methods section.

DEG criteria

We have now specified thresholds, adjusted p-values, and the Benjamini–Hochberg method for multiple testing. This information is added.

GEO processing

Platform IDs and normalization approach covariates have been described in detail. Implemented in Methods.

Tools/versions

Exact software, versions, and packages (where applicable) used for analysis and data processing have been added to ensure reproducibility. Incorporated in Methods.

Justification for PDE1A selection

We expanded the explanation for prioritizing PDE1A among the seven DEGs based on functional importance, enrichment relevance, and reported links with fibrosis. Added where applicable.

Pose validation

Pose clustering method, key residues, and selection criteria have been added. Incorporated at appropriate position in the manuscript.

Functional Enrichment

We now specify ontology categories (GO BP/CC/MF), KEGG and Reactome usage, significance cutoffs, background gene set, and multiple-testing corrections. Included in manuscript.

Statistics

Sample sizes, test types, normality checks, and two-sided assumptions are now clearly stated. Incorporated in manuscript.

Reviewer Comment: Some references were not aligned and formatting needed correction.

Response: All references have been rechecked and reformatted.

Reviewer Comment: Docking alone is insufficient; MD simulations must be performed and reported.

Response: We fully agree. We have now performed molecular dynamics simulations using GROMACS and Ligand RMSD, Protein RMSF, Hydrogen-bond and hydrophobic interaction occupancies, Water-bridge analysis, MM/PBSA binding energies (mean ± SD) and other necessary analyses have been performed. All MD methodology and results have been integrated into the manuscript, including new figures and tables.

Regulatory approval

We have now specified the agency and year of pirfenidone approval (FDA 2014; EMA 2011) with primary citations.

MD integration

The MD findings are now integrated at appropriate positions in the manuscript.

We have added PDB IDs and UniProt IDs for PDE1A, GEO dataset IDs with GSM groupings, platform IDs, and Links of the utilized public repositories.

All identifiers are now clearly listed in the manuscript.

Response: In compliance with the reviewer’s request, we have explained the GEO preprocessing docking protocols with tool names, parameters and versions; analyze MD trajectories and mentioned the results in the manuscript at appropriate positions, topologies, and analysis code. DOI included in manuscript where needed. This ensures full transparency and reproducibility.

All requested analyses have been performed and added, including RMSD, RMSF, MM/PBSA binding energies, and other necessary analyses.

These results are presented in the revised Results.

Reviewer #3 Comment:

The RMSF analysis could be strengthened by a residue-specific interpretation and quantitative comparison between apo and ligand-bound systems.

Response: The RMSF analysis has been revised to include residue-specific interpretation of regions 260–269, 330–333, 366–367, and 384–388, with explicit discussion of their secondary structure context and proximity to the binding pocket. Additionally, a comparison of RMSF values between apo and pirfenidone-bound systems performed and reduced fluctuation was observed in pirfenidone-bound systems. These revisions clarify the ligand-induced stabilization of PDE1A dynamics without requiring additional simulations, as suggested. The results have been mentioned in the manuscript at appropriate positions.

We are grateful for the reviewer’s detailed and rigorous comments. All recommended revisions have been thoroughly implemented, and the manuscript has been significantly strengthened in terms of methodology, reproducibility, scientific depth, and clarity. We have incorporated each modification into the revised manuscript as requested.

---

## [Decision Letter · Decision Letter 4]

26 Jan 2026

Dear Dr. Ullah,

plosone@plos.org . . A letter that responds to each point raised by the academic editor and reviewer(s). You should upload this letter as a separate file labeled 'Response to Reviewers'.A marked-up copy of your manuscript that highlights changes made to the original version. You should upload this as a separate file labeled 'Revised Manuscript with Track Changes'.An unmarked version of your revised paper without tracked changes. You should upload this as a separate file labeled 'Manuscript'.

We look forward to receiving your revised manuscript.

Kind regards,

Mahbub Hasan, PhD

Academic Editor

PLOS One

Journal Requirements:

Reviewers' comments:

Reviewer's Responses to Questions

**Comments to the Author**

Reviewer #3: All comments have been addressed

2. Is the manuscript technically sound, and do the data support the conclusions?

Reviewer #3: Partly

3. Has the statistical analysis been performed appropriately and rigorously?

Reviewer #3: No

4. Have the authors made all data underlying the findings in their manuscript fully available?

Reviewer #3: No

5. Is the manuscript presented in an intelligible fashion and written in standard English?

Reviewer #3: No

Reviewer #3: The manuscript can be accepted once all trajectory data and the code used to generate the related figures are made publicly available. For example, the complete trajectory dataset may be uploaded to Zenodo or a similar repository, with the corresponding access link included in the Data Availability section.

**Do you want your identity to be public for this peer review?** For information about this choice, including consent withdrawal, please see our For information about this choice, including consent withdrawal, please see our Privacy Policy .

Reviewer #3: No

---

## [Author Response · Author response to Decision Letter 5]

28 Jan 2026

Response to Reviewer Comments- R5

We are very grateful for the valuable time and constructive comments provided by both the editor and the reviewers. All observations have significantly improved the quality, clarity, and reproducibility of our manuscript. We have carefully addressed all reviewers’ comments and made every effort to incorporate all recommended changes.

Reviewer Comment: The manuscript can be accepted once all trajectory data and the code used to generate the related figures are made publicly available. For example, the complete trajectory dataset may be uploaded to Zenodo or a similar repository, with the corresponding access link included in the Data Availability section.

Response: We thank the reviewer for this valuable suggestion. In response, all trajectory data and the complete code used to generate the related figures have now been made publicly available. The full dataset has been deposited in the Zenodo repository and can be accessed via the following DOI: https://doi.org/10.5281/zenodo.18387883. The corresponding access link has also been included in the Data Availability section of the online portal as well as revised manuscript.

---

## [Editor Report · Decision Letter 5]

1 Feb 2026

Network Pharmacology and Integrative Bioinformatics analyses identify PDE1A as a key target of Pirfenidone in Idiopathic Pulmonary Fibrosis

PONE-D-24-60633R5

Dear Dr. Ullah,

We’re pleased to inform you that your manuscript has been judged scientifically suitable for publication and will be formally accepted for publication once it meets all outstanding technical requirements.

Kind regards,

Mahbub Hasan, PhD

Academic Editor

PLOS One
---

## [Editor Report · Acceptance letter]

PONE-D-24-60633R5

PLOS One

Dear Dr. Ullah,

I'm pleased to inform you that your manuscript has been deemed suitable for publication in PLOS One. Congratulations! Your manuscript is now being handed over to our production team.

Kind regards,

on behalf of

Dr. Mahbub Hasan

Academic Editor

PLOS One